# PEDV: Insights and Advances into Types, Function, Structure, and Receptor Recognition

**DOI:** 10.3390/v14081744

**Published:** 2022-08-09

**Authors:** Feng Lin, Huanyu Zhang, Linquan Li, Yang Yang, Xiaodong Zou, Jiahuan Chen, Xiaochun Tang

**Affiliations:** 1College of Animal Sciences, Jilin University, Changchun 130062, China; 2Chongqing Research Institute, Jilin University, Chongqing 401120, China

**Keywords:** porcine epidemic diarrhea virus (PEDV), PEDV variant strain, spike protein (S protein), PEDV receptor

## Abstract

Porcine epidemic diarrhea virus (PEDV) has been endemic in most parts of the world since its emergence in the 1970s. It infects the small intestine and intestinal villous cells, spreads rapidly, and causes infectious intestinal disease characterized by vomiting, diarrhea, and dehydration, leading to high mortality in newborn piglets and causing massive economic losses to the pig industry. The entry of PEDV into cells is mediated by the binding of its spike protein (S protein) to a host cell receptor. Here, we review the structure of PEDV, its strains, and the structure and function of the S protein shared by coronaviruses, and summarize the progress of research on possible host cell receptors since the discovery of PEDV.

## 1. Introduction

Porcine epidemic diarrhea virus (PEDV), a member of the genus Alphacoronavirus within the family Coronaviridae [1,2], was first reported in the United Kingdom [3] (1977) and Belgium [1] (1978). Although PEDV first appeared in Europe, since its emergence, it has been rampant in many countries in Asia and the Americas, including China, the United States, South Korea, Japan, and others. After a long period of wide-ranging transmission, PEDV has evolved mutant strains that display stronger virulence, infectivity, and pathogenicity, causing tremendous trouble for the pig industry worldwide. Therefore, research on the receptors that allow PEDV to enter mammalian cells is essential, and some progress has been made in this area.

## 2. An Overview of the Genome of PEDV and the Proteins It Encodes

PEDV is a positive-sense, single-stranded RNA-encapsulated virus with a single-stranded, positive-sense RNA genome (excluding polyA) of approximately 28 kb. The genome contains, in order, a 5′ end, a 5′ end cap, a 5′ untranslated region (UTR), 7 open reading frames, a 3′ untranslated region (UTR), and a polyadenylated (polyA) tail at the 3′ end [4]. The seven open reading frames encode four structural proteins: the S protein (spike protein), the M protein (membrane protein), the E protein (envelope protein), and the N protein (nucleocapsid protein), as well as the pp1a, pp1ab, and ORF3 proteins [5] (Figure 1).

The S protein is crucial for the entry of viruses into host cells. During cell entry, the S protein binds to receptors on the cell and mediates the subsequent fusion of the virus with the cell membrane [6,7]. The M protein is part of the viral envelope and participates in the assembly and release of viral particles [8,9,10]. The N protein is highly conserved and plays an important role in viral replication and transcription. In addition, the N protein and the viral RNA genome form a complex that serves as the core of PEDV. Although the E protein has not been studied intensively, data have demonstrated that the core composed of the N protein and genome is wrapped inside the virus under the joint action of the M and E proteins [11,12,13]. Moreover, the E protein affects the immune response of the host and thereby regulates PEDV infection [14,15]. ORF1a and ORF1b encode two polyproteins, pp1a and pp1ab, which in turn are processed into 16 nsps that play a role in the proliferation and synthesis of the viral genome. ORF3 acts as a coprotein and is involved in viral infection, with ORF3 deficiency decreasing virulence [16,17].

## 3. Types of PEDV Strains

Since the emergence of PEDV in the 1970s, many changes have occurred in its transmission process, resulting in the existence of many strains with different degrees of virulence and infectivity. Many scientists have analyzed the evolution and genotype of PEDV. The virus is usually classified into classical strains (GI) and mutant strains (GII), among them, mutant strains (GII) can be subdivided into non-S INDEL and S INDEL

### 3.1. Classic Strains (GⅠ)

The classical strains include early PEDV strains found in Europe and Belgium (virulent strain CV777) as well as some common strains and most cell culture-adapted mutant strains obtained by successive in vitro passaging, such as attenuated CV777 and DR13 [18]. The classical strains are less virulent than others, and although they have appeared in most regions of Europe and Asia, they primarily cause sporadic outbreaks [19]. Due to vaccination, the classical strains largely disappeared by 2010.

### 3.2. Mutant Strains (GII)

The mutant PEDV strains that emerged after 2010 have been divided into two categories: non-S INDEL PEDV and S INDEL PEDV.

#### 3.2.1. Non-S INDEL

Since 2010, there has been another large-scale PED epidemic in China, and the vaccine against the classical strain was unable to prevent the occurrence of a rampant epidemic, indicating the emergence of mutant PEDV strains with high virulence. Emerging strains of PEDV cause up to 100% morbidity and 80–100% mortality in piglets [20]. In June 2013, a highly virulent PEDV mutant also emerged for the first time in the United States [21]. Mutant strains have caused massive outbreaks in the Americas, China, Korea, Japan, and the Philippines because of their high infectivity and virulence [22,23].

#### 3.2.2. S INDEL

In 2011, W. Li et al. identified new mutant strains of PEDV in China [24]. Then, in 2014, Wang et al. identified another PEDV mutant strain, OH851, in Ohio, USA [25]. In the same year, Vlasova et al. sequenced and analyzed the genomes of 74 PEDV strains isolated in North America and found that the S gene of the emerging PEDV mutant strains contained insertions and deletions [26], and that this change made the emerging PEDV strains less virulent and less pathogenic. For this reason, these emerging strains and their analogs were designated “S INDEL PEDV” to distinguish them from the previous mutant strains (the “non-S INDEL PEDV” mentioned above). This distinction of the mutant PEDVs allowed its division into two categories: non-S INDEL PEDV and S INDEL PEDV.

In 2015, Lee et al. analyzed the genome-wide phylogeny of PEDV worldwide. In 2019, Guo et al. compared the genome of S INDEL PEDV with that of previous PEDV strains using SimPlot v.3.5.1, a form of sequence similarity mapping software. Both groups found that S INDEL PEDV may have arisen as a result of a recombination event between classical strains and mutant strains [22,23]. The S protein of S INDEL PEDV has higher similarity with that of the classical strain. The virulence and pathogenicity of the classical strain are lower than those of non-S INDEL PEDV, and these differences may be due to this distinction, as the virulence and pathogenicity of S INDEL PEDV are far less than those of non-S INDEL PEDV.

We compared the amino acid sequences of the S proteins of 53 different PEDV strains (15 GⅠ strains and 38 GⅡ strains; of the latter, 31 were non-S INDEL strains and 7 were S INDEL strains) (Table 1). Our analysis showed that the S proteins of GⅠ strains and GⅡ non-S INDEL strains differed greatly from each other, with a similarity rate of only 90–94%. The differences were concentrated in the N-terminal structural domain, indicating that the N-terminal region of S1 showed a high genetic diversity among PEDV strains, while the similarity in the S proteins encoded by GⅡ S INDEL strains and GⅠ strains was relatively high, reaching approximately 96%. We then focused on a comparison of the N-terminal ends of the S proteins of PEDV strains and found a large number of fragment deletions and insertions (Figure 2).

## 4. Life Cycle of PEDV

The binding of PEDV to the surface receptors of host cells initiates the life cycle of PEDV (Figure 3), consistent with the classical coronavirus life cycle [27,28]. The main steps are as follows:

PEDV binds to the host cell receptor and adsorbs to the surface of the cells. Its genome is either directly released into the cells through membrane fusion [23] or the virus is engulfed by the cells to form an endosome, allowing it to enter the cell and release its genome [29].

The PEDV genome is directly translated into nsp1-16 (including RTC) after binding to cellular ribosomes.

Viral mRNA biosynthesis and genome replication occur through discontinuous synthesis [30,31,32].

Discontinuous genomes are translated into structural proteins: the S protein, the E protein, the M protein, the N protein, and auxiliary protein.

The S, E, and M proteins are inserted into the endoplasmic reticulum (ER) membrane and then transferred to the ER–Golgi intermediate compartment (ERGIC). The viral genome bound to the N protein is assembled and enters the ERGIC membrane containing the viral structural proteins, resulting in the formation of mature virosomes.

Finally, the virus is released outside the cell, where it begins a new round of infection [33].

The PEDV life cycle begins with the binding of the S protein to the host cell through interaction with its receptor. Upon viral entry, the PEDV genome is released and the two large open reading frames, ORF1a and ORF1b, are immediately translated, resulting in the production of polyproteins pp1a and pp1ab. These two polyproteins are cleaved into 16 nsps, including the replication and transcription complex (RTC), which then participates in viral genomic RNA replication and subgenomic mRNA (sg mRNA) transcription, a process that occurs in a microenvironment consisting of perinuclear double-membrane vesicles (DMVs), coiled membranes (CMs), and small open double-membrane microspheres (DMSs) (not shown in Figure 3). The viral structural protein N and the viral genomic RNA form a helical nucleocapsid, and structural proteins M, S, and E translocate into the endoplasmic reticulum (ER), from which viral particles containing the helical nucleocapsid are subsequently assembled in the ERGIC. Finally, the viral particle is secreted from the infected cell by cytosolic exocytosis. 

## 5. Key to the Entry of PEDV into Target Cells—The S Protein

Under normal circumstances, the cell membrane protects the cell from viruses by acting as a screen. The virus cannot enter the cell unless it matches the lock present on the host cell—the receptor on the surface of the cell membrane. To understand how PEDV interacts with its receptor, we must understand the PEDV key.

### 5.1. Structure of the Coronavirus S Protein

Viral fusion proteins have been divided into two categories based on their structural similarities: type I viral fusion proteins and type II viral fusion proteins. Whereas type I viral fusion proteins contain a heptad repeat (HR) region and a fusion peptide (FP) at or near the N-terminus of the S2 subunit, type II viral fusion proteins lack an HR region but possess an internal FP. The coronavirus S protein belongs to the type I viral fusion protein family; it contains an internal FP with two hydrophobic HR regions, referred to as HR1 and HR2 [6].

Some coronaviruses have been studied by cryo-electron microscopy (cryo-EM). These include betacoronaviruses such as SARS-CoV, mouse hepatitis virus (MHV), MERS-CoV, HKU1, and SARS-CoV-2 [34,35,36,37,38], and a small number of alphacoronaviruses, including 229E, NL63, and PEDV [39,40,41]. Studies have shown that the S protein contains an S1 subunit that participates in receptor binding and an S2 subunit that participates in membrane fusion. The cryo-EM structure of PEDV has also been solved in recent years; a schematic diagram of PEDV based on its cryo-EM structure is shown in Figure 4.

#### 5.1.1. S1 Subunit

Cryo-EM studies of the genus Betacoronavirus have revealed that the S1 subunit of the S protein folds into four separate structural domains, which some scientists have designated Domain A, Domain B, Domain C, and Domain D. Others have called these domains NTD, CTD, SD1 (subdomain 1), and SD2 (subdomain 2) [42,43,44].

Betacoronavirus S1 subunits contain four individually folded domains: domains A-D. Domain A (NTD) and domain B (CTD) are related to receptor binding, and protein receptors usually bind viruses through domain B (CTD) of S1, while glycans bind viruses through domain A (NTD) [45].

An important feature of the coronavirus S1 subunit is its sequence diversity and the variability in the way in which it binds to receptors. Although all viral S1 subunits share a common evolutionary origin, different sequences and structures of the S1 subunit have evolved after frequent cross-species transmission and recombination, and viral S1 subunits thus have the ability to bind to a variety of receptors [46]. Studies have shown that the S1 NTD of coronaviruses is responsible for binding glycans; this occurs in the betacoronaviruses hCoV-OC43, hCoV-Huh1, and MERS-CoV, and the alphacoronaviruses TGEV and PEDV.

As early as the twentieth century, 9-O-acetylated sialic acid (9-O-Ac-Sia) was found to be necessary for the entry of HCoV-OC43 into cells [47]. In 2015, Huang et al. demonstrated that the S1 structural domain of hCoV-HKU1 can bind to RD cells (human rhabdomyosarcoma/muscle tumor cells), that this binding is sialic acid-dependent, and that neuraminidase (NA) pretreatment reduced the binding of HKU1-S1 to RD cells [48]. A few years later, Hulswit et al. concluded that HKU1 S1 domain A binds to 9-O-Ac-Sia in a 9-O-Ac-dependent fashion through a low-affinity, high-specificity interaction. Detection of this interaction is not easy. This may also explain why preincubation of cells with soluble S1 domain A–Fc fusion proteins did not block HKU1 infection [49]. W. Li et al. found that the MERS-CoV S protein binds sialic acid, that the sialic acid-binding site is located within domain A (aa 18–355) of the MERS-CoV S1 subunit, and that sialic acid acts as an attachment receptor for MERS-CoV and contributes to virus entry into cells. W. Li et al. infected Calu-3 cells with MERS-CoV and found that the rate of infection of Calu-3 cells by the virus was reduced by more than 70% after NA treatment [50]. The TGEV S protein of the genus Alphacoronavirus recognizes N-ethyleneglycolyl neuraminic acid (Neu5Gc) and N-acetylneuraminic acid (Neu5Ac), and this glycan-binding activity is required for intestinal homing of TGEV [51].

F. Li et al. compared the S1 subunits of coronaviruses in different genera and found that they have low similarity, while those of species within the same genus have high sequence similarity [46]. Recently, Walls et al. observed the HCoV-NL63S protein by cryo-EM and compared alphacoronaviruses with betacoronaviruses in phylogenetic analyses. The S1 subunits of most alphacoronaviruses (including HCoV-NL63) contain an additional domain at the N-terminus of the subunit relative to the betacoronavirus S1 subunit; this additional domain is produced by NTD replication, and due to its similarity to the N-terminus of the NTD, Walls’ group called it Domain 0. The structure of domain 0 is similar to that of the sialic acid-binding domain of the VP4 S protein of rotavirus. The authors speculated that this domain plays a role in binding to sialic acid [40].

In 2019, Wrapp et al. observed the fusion preconformation of PEDV S protein by cryo-EM [41] and found that the protein consisted of an S1 subunit (aa 1–755) and an S2 subunit (aa 756–1383). There is a domain (aa 34–234) at the N-terminus of S1-NTD that is similar to domain 0 of HCoV-NL63; this domain is referred to as DΦ by the authors. Determination of the structure of PEDV S protein verified the results of the phylogenetic analysis performed by Walls et al., which showed that alphacoronavirus S protein contains an additional domain at the N-terminus relative to betacoronavirus S protein. Current studies have demonstrated that PEDV binds to sialic acid via the structural domain DΦ, as described in a later section.

The S1 CTD of coronavirus is responsible for binding to protein receptors on cells. Among alphacoronaviruses, hCoV-NL63 utilizes hACE2 as a receptor, hCoV-229E and TGEV bind to APN to enter host cells, and PEDV may also use APN as a receptor. The betacoronaviruses SARS-CoV and SARS-CoV-2 use hACE2 as a receptor, similar to hCoV-NL63, and the receptor for MERS-CoV is DPP4 [38,52,53,54,55,56]. All of the above coronaviruses bind carbohydrates through the S1 NTD and proteins through the S1 CTD; the only known exception is the betacoronavirus MHV, which binds to the MNV protein receptor CEACAM1 via the S1 NTD.

Almost all coronaviruses rely on binding of the RBD of S protein to protein receptors to enter the host cell. The RBDs of the S proteins of all coronaviruses for which this has been determined to date, except MHV, are located on the CTD of the S1 subunit (Table 2). The exact location of the RBD of PEDV has not been determined, but pAPN, a possible receptor for PEDV, can bind to the S1 CTD (aa 477–629) of the S protein; thus, the RBD of PEDV may be located in the S1 CTD between aa 477–629 [57].

#### 5.1.2. S2 Subunit

The amino acid sequence of the S2 subunit is more conserved than that of the S1 subunit. Similar to class I fusion proteins, all coronavirus S2 subunits contain four elements that are required for membrane fusion: the FP, two HR regions (HR1 and HR2), and a transmembrane domain (TM) [6].

The FP is highly conserved and is present in all coronavirus S proteins. It is a hydrophobic peptide in which residues 15–25 mediate fusion between the virus and the cell membrane. The FP of PEDV is located in the region formed by aa residues 892–910, which forms an α-helix at the N-terminus of the S2′ protease cleavage site [41,42,63]. The six-helix bundle structure formed by HRs (HR1 and HR2) during membrane fusion and the antiparallel six-helix bundle composed of HR1 and HR2 are the unifying features of class I virus fusion proteins. This six-helix bundle structure helps the FP insert into the host cell membrane and mediate membrane fusion [31,38,64].

### 5.2. Protease-Mediated Fusion Activation of Coronavirus S Protein

The S protein of coronavirus contains at least two protease cleavage sites, namely, the S1/S2 site, which is located between the S1 and S2 subunits, and the S2′ site, which is located at the N-terminus of the FP. Both sites are cleaved by a host protease as a prerequisite for the entry of coronavirus into the host cell [65].

Wrapp et al. observed the structure of the S protein of PEDV and speculated that the S1/S2 site is located between Lys755 and Ser756, and that the S2′ site is located next to Arg891 [41]. For coronaviruses to enter their host cells, sequential cleavage of the S1/S2 site and the S2′ site is essential. The S proteins of SARS-CoV [66], MERS-CoV [67], and SARS-CoV-2 [38] are all hydrolyzed in the order of the S1/S2 site and then the S2′.

Coronavirus infection is mediated through viral surface glycoproteins that consist of S1 and S2 subunits and form trimer spikes on the envelope of the virion. In the prefusion trimer spike formed by the S protein, the S2 subunit is embedded under the S1 subunit, and the protein is not easily cleaved. When the S1/S2 site is cleaved, the S2′ site on the S2 subunit becomes exposed, making it easy to cleave. Therefore, cleavage of the S1/S2 site promotes S2′ cleavage [65,68]. Since coronavirus membrane fusion is a relatively conserved process, it is speculated that PEDV may also be hydrolyzed in a similar way through two-step cleavage of the S protein. The three most common host cell proteases that activate coronavirus S protein are trypsin, furin, and cathepsin L. Furin protease is a representative member of the preprotein convertase family. It can cleave the S1/S2 and S2′ sites via a furin cleavage recognition motif. The minimum recognition motif of furin is -Arg-X-X-Arg ↓—(X: any amino acid; ↓: the cleavage site), and whether cleavage can occur is related to the variation of the motif. For example, furin protease can also recognize Lys/Arg-X-X-Lys/Arg-Arg ↓- [69]. Most well-known furin cleavage sites of coronaviruses, such as those of HCoV-HKU1, MERS-CoV, SARS-CoV-2, and MHV, are located at S1/S2. However, there are some exceptions. It has been reported that some coronaviruses have furin cleavage sites at the S2′ site; one example is the IBV Beaudette strain, which can be cleaved at the S2′ site [70]. In addition, according to a bioinformatic analysis, MERS-CoV may also have a furin cleavage site at S2′ [38,67,71,72]. The furin cleavage site is important for coronaviruses, and the replication rate of SARS-CoV-2, which lacks the furin cleavage site, in human cell lines is low [73,74,75].

Cathepsin belongs to the class of lysosomal proteases that usually exist in lysosomes and endosomes [72]. It is also a common host protease related to the cleavage and activation of coronavirus S proteins, including those of SARS-CoV [76], MERS-CoV [77], and SARS-CoV-2 [38], the last of which has recently spread widely.

The role of trypsin in the activation of coronavirus glycoproteins by cleavage has long been known. Trypsin is not selective in its substrate recognition, and there are many possible trypsin cleavage sites in the coronavirus S protein. Trypsin has a strong preference for cleavage after arginine (R) or lysine (K) residues [72]. Previous studies have shown that exogenous trypsin enhances SARS-CoV S1/S2 hydrolysis and infection [78,79].

Millet et al. analyzed the amino acid sequences of some coronaviruses and found no furin cleavage site at the S1/S2 boundary or within the S2′ site of PEDV [67]. Liu et al. used a pseudovirus encoding the PEDV S protein to illustrate that furin protease cannot activate spike proteolysis and subsequent entry of PEDV, but that it can activate the spike of a MERS-CoV pseudovirus that was used as a control [80]. In addition, W. Li et al. constructed a PEDV strain with a mutation in S2′ that introduced a furin cleavage site. Infection of Vero cells by wild-type PEDV depends on trypsin, but the mutant PEDV infected Vero cells in a trypsin-independent manner, and this infection was blocked by furin-specific inhibitors [81]. The above data confirm that there is no furin cleavage site in PEDV, and since infection of cells by wild-type PEDV requires trypsin, it is speculated that the S2′ site of PEDV S protein may be activated by trypsin.

The transition in the conformation of the S protein trimer of PEDV from the prefusion state to the postfusion state and separation of the S1 and S2 subunits may require the action of trypsin or a trypsin-like host protein. Studies illustrate that trypsin can facilitate PEDV infection of host cells in vitro [82,83,84,85]. Moreover, cathepsin is more effective in activating PEDV spike-mediated entry into cells than trypsin [80]. Trypsin, cathepsin, and other proteases that cleave PEDV spikes can be used in in vitro PEDV culture. However, it is not clear whether these proteases are necessary for PEDV spike cleavage in vivo or which protease or proteases act in the in vivo environment. Some scholars speculate that gastric and pancreatic proteases or proteases that are locally expressed in intestinal epithelial cells may promote these processes that are critical to PEDV infection of animal hosts. However, trypsin or cathepsin can clearly be used to hydrolyze PEDV spikes under in vitro conditions, and this facilitates the study of the interaction of PEDV with its host cells. Through continuous passage of PEDV in cells, mutant strains such as DR13, 83P-5, and SM98-1 can be obtained that are adapted to cell culture and that directly enter host cells and spread without a need for protease activity [18,83,86]. The emergence of mutant strains in vitro is helpful in the study of PEDV infection and in vaccine preparation.

### 5.3. S Protein-Mediated PEDV Entry into Host Cells

The prefusion and postfusion structures of many coronaviruses, including the alphacoronaviruses hCoV-229E [58], hCoV-NL63 [40], and PEDV [41], and the betacoronaviruses SARS-CoV [43], MERS-CoV [44,87], hCoV-HUK1 [37], and SARS-CoV-2 [38], have been observed by cryo-EM. The results show that the membrane fusion processes of these coronaviruses are similar.

Cryo-EM studies reveal similar conformations of the coronavirus S protein at various stages of virus–host cell fusion. Prior to fusion of the virus with the host cell, the S1 subunit wraps the S2 subunit tightly, and the RBD on the S1 subunit is buried within the trimer so that the host cell receptor cannot bind to the RBD; the RBD at this point is referred to as being in the “down” conformation. In the prefusion state of the S protein, the S1 CTD is no longer buried but becomes exposed, and the RBD must flip upward to bind to the receptor in the “on” conformation, which facilitates the binding of the RBD to the viral receptor. Binding of the RBD to viral receptors leads to activation of the S protein trimer. Host proteases further act on the S2′ site at the N-terminus of the FP on the S2 subunit so that the FP is exposed and inserts into the host cell membrane. At the same time, HR1 and HR2 on the S2 subunit fold to form a six-helix beam structure in such a way that the FP and the TM are aligned, causing the virus and the cell membrane to fuse; the genome then enters the host cell.

### 5.4. The S Protein Is the Key Determinant of Viral Host Range

Receptor recognition is a key determinant of viral host range and cellular tropism. The spike of coronavirus is the key that allows it to lock onto its host cells. For example, SARS-CoV is a zoonotic pathogen derived from animals, and civets are considered an important intermediate host in its transmission to humans [88]. Graham et al. compared the RBD sequences of SARS-CoV isolated from humans and civets and found six residues that differ between them **[89]**. W. Li et al. isolated SARS-CoV S proteins from civets and found that those that possessed K479 and S487 residues had high affinity for civet ACE2, while their affinity for human ACE2 was much lower. The affinity of S proteins with K479N and S487T mutations for human ACE2 was increased, as was the ability of the virus carrying these mutations to infect human cells [90]. Other reports have confirmed that the K479N and S487T mutations are specifically involved in the transmission of SARS-CoV from civets to humans [91]. Thus, the S protein is one of the main barriers to SARS-CoV transmission between animals and humans [88,92].

A recent study of PEDV cellular tropism by Z. Li et al. demonstrated that the cellular tropism of PEDV depends on the S protein and on the interaction between the S1 and S2 subunits; the group identified an important role for two amino acid residues within the S protein, 803L and 976H, in PEDV cellular tropism [93]. 

## 6. Advances in PEDV Receptor Research

The receptors of most of the known coronaviruses in humans have been identified. PEDV belongs to the genus Alphacoronavirus, and numerous studies aimed at identifying its cellular receptor are still ongoing. Although the identity of the receptor is not conclusive, there have been many significant advances.

### 6.1. Aminopeptidase N

Aminopeptidase N (APN), which is commonly known in the field of immunology as astrocyte antigen CD13, is a type II transmembrane protein. It contains multiple coronavirus-binding sites, and different viruses bind at different sites for entry into host cells [94].

When looking for the PEDV receptor, scholars first focused on porcine aminopeptidase N (pAPN). There are two main reasons for this focus. First, as early as the end of the twentieth century, APN was identified as a receptor for TGEV, FCoV, canine coronavirus (CCV), and hCoV-229E [95,96,97]. Second, PEDV mainly infects the small intestine of pigs, which is rich in pAPN expression. This has led to the prioritization of investigation of APN as a possible PEDV receptor [98,99].

In 2003, Oh et al. speculated that pAPN might be a PEDV receptor based on the similarity of the structure of PEDV to that of HCoV-229E and TGEV, and verified that the interaction between PEDV and pAPN could be disrupted by an anti-pAPN antibody in swine testis (ST) cells and intestinal cells [100]. In 2007, B.X. Li et al. overexpressed pAPN in MDCK cells, making MDCK cells susceptible to PEDV [101]. Moreover, Nam et al. found that the ability of PEDV to infect ST cells is correlated with the density of pAPN on the cell membrane and that overexpression of exogenous pAPN in ST cells enhanced PEDV infection [86]. Numerous studies based on this observation followed. In 2015, Liu expressed the S1 RBD fragment and pAPN [102,103] and showed by dot blot hybridization that the PEDV S1 subunit binds effectively to pAPN. In addition, pseudovirus expressing the PEDV S protein could enter MDCK (canine kidney) cells overexpressing pAPN, but wild-type MDCK cells were not infected. These data confirm that pAPN can effectively bind to PEDV S protein and mediate PEDV entry into host cells.

pAPN contains seven structural domains (structural domains I–VII); domains I, II, and III are responsible for anchoring the protein at the cell surface. In 2017, Kamau et al. constructed two mutant NIH3T3 cell lines that overexpress pAPN: pAPNDI-VI (a deletion mutant lacking domain VII) and pAPNDVII (deletion mutant lacking domains IV through VI). NIH3T3 cells overexpressing pAPNDI-VI were not infected by PEDV, while wild-type NIH3T3 cells and pAPNDVII-overexpressing NIH3T3 cells remained susceptible to infection by PEDV [104].

The above studies appear to provide solid evidence that pAPN is the functional host receptor of PEDV. However, further research in recent years has not produced satisfactory results. This tells us that the PEDV receptor is far from being understood. In 2017, W. Li et al. used dot blot hybridization to search for PEDV receptors, as Liu did in 2015, but failed to find pAPN binding to PEDV spikes regardless of overexpression of pAPN or genetic ablation in MDCK, ST, Huh7, and HeLa cells. As a control, they showed that TGEV S protein binds to pAPN in MDCK and ST cells [105]. In 2018, Ji et al. reached a similar conclusion, namely, that APN did not affect PEDV infection in Vero cells or IPEC-J2 cells, based on overexpression and genetic ablation experiments [106]. The results of numerous studies have been inconsistent. Moreover, the in vitro conditions used in these studies do not necessarily replicate the natural host environment. Based on the above consideration, in 2019, the Yang and Prather groups developed ANPEP knockout pigs using the CRISPR/Cas9 system and evaluated the biological relevance of pAPN as a putative receptor of PEDV [107]. The results showed that *ANPEP*-null pigs retained susceptibility to PEDV infection, suggesting that APN is not the functional PEDV receptor [108,109,110]. These unexpected phenomena allow scientists to explore the reasons for these findings and thus put forward some conjectures.

In 2016, Shirato et al. hypothesized that porcine aminopeptidase N is not a cellular receptor for PEDV, but that it promotes PEDV infection through its aminopeptidase activity. Shirato constructed CPK-pAPN (wild type) and CPK-pAPNmt (pAPN mutant without APN enzyme activity) transiently transfected cell lines using CPK cells from porcine kidney. He found that PEDV replicated much more strongly in CPK-pAPN cells than in CPK-pAPNmt cells, indicating that pAPN activity may be involved in PEDV infection. Next, the authors used a pAPN protease activity inhibitor to verify this hypothesis, with the results showing that the replication of PEDV in CPK-pAPN cells was inhibited by the pAPN protease inhibitor. According to the experimental results, it could be concluded that APN is not the cell receptor of PEDV but that it promotes PEDV infectivity through the activity of the pAPN enzyme [111]. Although this conclusion has not been confirmed, it may provide some inspiration for subsequent studies of the role of APN in the propagation of PEDV.

### 6.2. Sialic Acid

Sialic acid (SA), also known as N-acetylneuraminic acid, is a naturally occurring carbohydrate. SA is attached to oligosaccharides, glycolipids, and glycoproteins, some of which act as receptors for viruses such as influenza, coronavirus, and rotavirus, allowing host entry [112,113].

Among coronaviruses, BCoV, HCoV-OC43, HEV, and MHV have O-acetylesterase activity that hydrolyzes sialic acid and removes receptor determinants from infected cells, thereby facilitating the spread of viruses. The removal of receptor determinants on the cell surface renders the cells resistant to virus aggregates, and this can be detected using hemagglutination assays. TGEV also has sialic acid-binding activity, and in addition to APN-binding activity, it possesses hemagglutinating activity but does not contain acetylesterase activity. The sialic acid-binding activity of TGEV is located on the spike, and certain mutations abolish the ability to agglutinate erythrocytes. Although sialic acid binding is an accessory factor for TGEV infection, the enteropathogenicity of the virus is lost when the spike is mutated within residues 145 to 155 [59,114,115,116,117].

In 2015, Liu et al. showed that the PEDV S1-NTD-CTD fragment binds to porcine and bovine mucins but not to porcine or human APN. The mucin binding of PEDV S1-NTD-CTD was reduced by incubation with neuraminidase, which hydrolyzes sialic acid. Testing against a glycan screen array containing 609 natural or synthetic mammalian glycans showed that PEDV preferentially binds Neu5Ac. The results demonstrated that PEDV employs both protein receptors and sugar coreceptors for infection. Utilization of Neu5Ac may explain the enteric tropism of PEDV because Neu5Ac is also a receptor determinant for TGEV [118,119].

In 2016, W. Li et al. further analyzed the location of sialic acid binding within the spike of PEDV and showed that the S1 NTD (residues 1-249) of the GDU strain binds to sialic acid, as verified by the interaction of PEDV and S1 protein with erythrocytes. The N-terminal 249 residues of S1 bind sialic acid and are able to agglutinate erythrocytes and facilitate host cell entry. Structurally, although Wrapp et al. found that the sialic acid-binding domain at DΦ has a different orientation from that of HCoV-NL63, sialic acid does not influence the prefusion coronavirus spike conformation [41].

Despite the importance of sialic acid interaction with PEDV for cell attachment and entry, in vitro and in vivo data suggest that sialic acid binding is not essential for host cell infection and that sialic acid may be a coreceptor of PEDV.

It has been found that different PEDV strains have different abilities to bind sialic acid; this may be caused by mutations in the PEDV S protein, most of which occur in the N-terminal domain [120,121].

In 2016, W. Li et al. found differences in the sialic acid-binding abilities of different PEDV strains. In a comparison of strains DR13, GDU, and UU, the sialic acid-binding ability of strain GDU was found to be much stronger than that of strains DR13 and UU. Li’s group also analyzed the amino acid sequences of the S1 subunits of different PEDV strains and found 69 amino acid residue differences, of which 51 were located in the N-terminal domain of S1 (aa 1–249) [41,64].

In 2016, Deng et al. compared the S protein mucoprotein-binding characteristics of two PEDV strains and found that the classical strain (CV777) showed much weaker mucoprotein binding than the emerging mutant strain (CHGD-01). The group purified and expressed the N-terminal structural domains of CV777 S1 (aa 1–320) and CHGD-01 S1 (aa 1-324) and used ELISA to detect the mucoprotein-binding activity of the S1-NTDs of the two different strains; the results showed that the mucoprotein-binding ability of the CV777 S1-NTD is much weaker than that of the CHGD-01 S1-NTD [57].

TGEV and PEDV are very similar; both are alphacoronaviruses, both infect the porcine intestine, and TGEV binds to sialic acid. The latter feature may promote the binding of TGEV to mucin on the surfaces of intestinal epithelial cells, enhancing its intestinal pathogenicity [117]. The emerging mutant PEDV strains carry S protein mutations that are mostly concentrated in the structural domain within the N-terminus, and their virulence and pathogenicity are much higher than those of the classical strain. The potential receptor or coreceptor sialic acid-binding activity may play an important role in the increased virulence and pathogenicity of these strains. This possibility deserves further reflection and study.

Currently, it is known that the ability to bind sialic acid varies among strains. Although at the present time there is no evidence that mutations in S proteins affect their binding to their receptors, this remains an effect that must be considered.

### 6.3. DC-SIGN/L-SIGN

DC-SIGN (CD209) and its homolog L-SIGN (also known as DC-SIGN-R or CD209 L) belong to the C-type (calcium-dependent) lectin family. Many viruses, including coronaviruses such as SARS-CoV and SARS-CoV-2, have been found to use DC-SIGN/L-SIGN to enter and infect host cells [122].

Han et al. conducted experiments using HeLa cells that were not infected with SARS-CoV and found that SARS-CoV pseudovirus could infect cells transfected with ACE2, DC-SIGN, or L-SIGN plasmids but not control cells that had been transfected with empty vectors, suggesting that DC-SIGN/L-SIGN can mediate SARS-CoV infection independently of ACE2. Further analysis revealed that cotransfection with DC-SIGN/L-SIGN and ACE2 plasmids enhanced SARS-CoV pseudovirus infection, suggesting that DC-SIGN/L-SIGN may be cofactors of ACE2 [123].

The effect of DC-SIGN/L-SIGN on SARS-CoV-2 infection is similar to its effect on SARS-CoV infection. Recently, Lu et al. found that SARS-CoV-2 S protein pseudovirus could infect HEK293T cells overexpressing ACE2 or DC-SIGN/L-SIGN but not controls transfected with an empty vector, indicating that DC-SIGN/L-SIGN binds directly to SARS-CoV-2 S protein and mediates the attachment of pseudovirus to cells. Subsequently, Lu’s group found that DC-SIGN/L-SIGN interacts with the S1 subunit of SARS-CoV-2 but largely did not bind to the RBD fragment, suggesting that DC-SIGN/L-SIGN may bind the S protein through regions other than the RBD [124].

Zhao et al. used BHK (baby hamster kidney) cells and NIH3T3 cells (mouse fibroblasts) that were not infected with PEDV to perform experiments. Transient expression of hDC-SIGN, hL-SIGN, or pDC-SIGN made BHK cells susceptible to infection with PEDV. Mannose and specific antibodies blocked the action of C-type lectins and reduced infection of hDC-SIGN-, hL-SIGN-, or pDC-SIGN-overexpressing cells by recombinant PEDV-GFP, demonstrating that hDC-SIGN/L-SIGN or pDC-SIGN can mediate the cell entry and dissemination of PEDV. However, the role of DC-SIGN/L-SIGN in PEDV infection and whether pDC-SIGN is the true receptor for PEDV have not yet been determined [125]. 

### 6.4. Heparan Sulfate

Heparan sulfate (HS) is a cell adhesion receptor that mediates the attachment and internalization of viruses and is expressed on the surfaces of almost all cells. HS is a prerequisite for the entry and infection of host cells by a variety of coronaviruses, including HCoV-NL63 and SARS-CoV-2 [126]. Clausen et al. not only demonstrated that HS is an essential host attachment factor that promotes SARS-CoV-2 infection in various target cells, but also found that the S protein interacts with HS via its RBD (aa 319–529) and that this interaction triggers a conformational change in the S protein RBD that facilitates the binding of the virus to its specific receptor (ACE2) [127].

In 2015, Huan et al. found that the attachment of PEDV to Vero cells required the presence of HS on the cell surface and that the addition of heparin (an HS analog) resulted in significant dose-dependent inhibition of PEDV binding to cells; moreover, removal of HS using heparinase I (an enzyme that can digest HS) and reduction of HS sulfation using sodium chlorate both reduced PEDV infection, indicating a requirement for acetyl heparin sulfate for virus attachment [128]. In 2020, Zhou et al. reported the antiviral effect of an HS analog, mercaptosulfate (MES), on PEDV [129].

### 6.5. Other Auxiliary Receptors in PEDV Infection

#### 6.5.1. Occludin

Occludin is a tight junction protein that is widely present in epithelial cells and endothelial cells and attaches adjacent cells to each other, forming intercellular seals. Occludin has been identified as a necessary host factor for the entry of several viruses, including coxsackie B virus (CBV), rotavirus, and human hepatitis C virus (HCV) [130].

Luo et al. overexpressed occludin in Vero E6 and IPEC-J2 cells and found that this greatly enhanced PEDV infection, but overexpression of occludin in BHK cells did not make the cells susceptible, suggesting that occludin alone was not sufficient to make the cells susceptible but could enhance PEDV infection. Small interfering RNA (siRNA) was used to inhibit endogenous occludin expression in Vero E6 and IPEC-J2 cells, and it was found that knockdown of occludin inhibited PEDV infection, demonstrating that occludin has an enhancing effect on PEDV infection [131]. Although occludin plays an important role in the process of PEDV infection, it is not clear whether occludin binds directly to PEDV, nor has the specific mechanism of action of occludin been determined.

#### 6.5.2. Integrin αvβ3

Integrins are cell surface heterodimeric glycoproteins containing α and β subunits. Integrin αvβ3 consists of αv and β3 subunits that have been shown to act as entry receptors for a variety of viruses, including herpesviruses, hantaviruses, and rotaviruses [132,133].

The binding of integrin to coronavirus S proteins is accomplished through recognition of short amino acid sequences on the protein. Sun et al. analyzed the integrin recognition sequences of PEDV S proteins and found four conserved integrin recognition amino acid motifs (Asp-Gly-Glu, Lys-Gly-Glu, Arg-Leu-Asp, and Leu-Asp-Val) in the S proteins of various PEDV strains, suggesting that integrin proteins may act as infection-related proteins for the attachment and entry of PEDV [134].

In 2017, C. Li et al. overexpressed integrin αvβ3 in Vero E6 cells and porcine intestinal epithelial cells (IECs) and found that this enhanced infection of the cells by PEDV CV777 and HM2017. In addition, coexpression of pAPN and integrin αvβ3 in Vero E6 cells significantly enhanced infection by PEDV strain CV777, indicating that integrin αvβ3 plays an important role in PEDV infection and that integrin αvβ3 and pAPN have synergistic effects on the enhancement of PEDV replication. Integrin αvβ3 miRNAs inhibited infection by the CV777 strain. The Arg-Gly-Asp (RGD) amino acid sequence is the major attachment site through which integrin αvβ3 interacts with proteins, and RGD peptides competitively and significantly inhibited replication of the CV777 strain in Vero E6 cells. However, the investigators found that HeLa and BHK-21 cells overexpressing integrin αvβ3 were not infected by PEDV, suggesting that integrin αvβ3 is not a functional receptor for PEDV but that it may function as a coreceptor [135].

## 7. Conclusions

In recent decades, PEDV has become a serious threat to the pig industry, causing massive economic losses. In previous studies, through the efforts of scientists, progress has been made in understanding the receptor mechanism of PEDV; for example, pAPN, sialic acid, DC-SIGN/L-SIGN, and HS have the potential to act as receptors or coreceptors, but it appears that much work must still be undertaken to fully reveal the receptor of PEDV. At present, due to the frequent mutation of PEDV and because research on the PEDV receptor has not been completed, the development of vaccines and drugs is also difficult, and there is no effective vaccine. The identification of PEDV receptors can not only provide more drug targets for the prevention and treatment of PEDV, but would also be helpful in the development of vaccines and would have far-reaching significance. Therefore, it is urgent to identify the PEDV receptor and to elucidate the mechanism of PEDV entry into host cells.

## Figures and Tables

**Figure 1 viruses-14-01744-f001:**
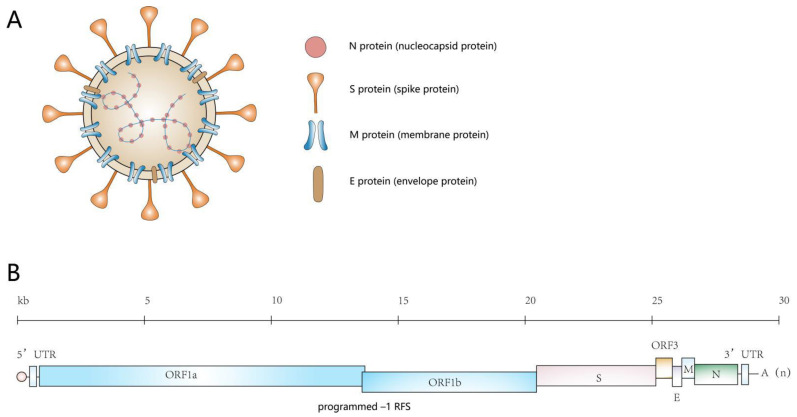
Schematic representation of PEDV genome organization and virion structure. (**A**) Schematic diagram showing the structure of the virus. The lipid bilayer containing the S, M, and E proteins covers the core structure of the viral RNA genome, which is bound to the N protein, forming a long helical ribonucleoprotein (RNP) complex. (**B**) Structure of the RNA genome of PEDV. The RNA genome of PEDV is 28 kb in size and has a 5′ end cap and a 3′ polyadenylated tail. The viral genome is flanked by UTRs and contains seven open reading frames: ORF1a, ORF1b, S, ORF3, E, M, and N; these are indicated by the boxes. (PEDV’s ORF1a and ORF1b have an overlapping region within which the programmed –1 ribosomal frameshift (RFS) occurs).

**Figure 2 viruses-14-01744-f002:**
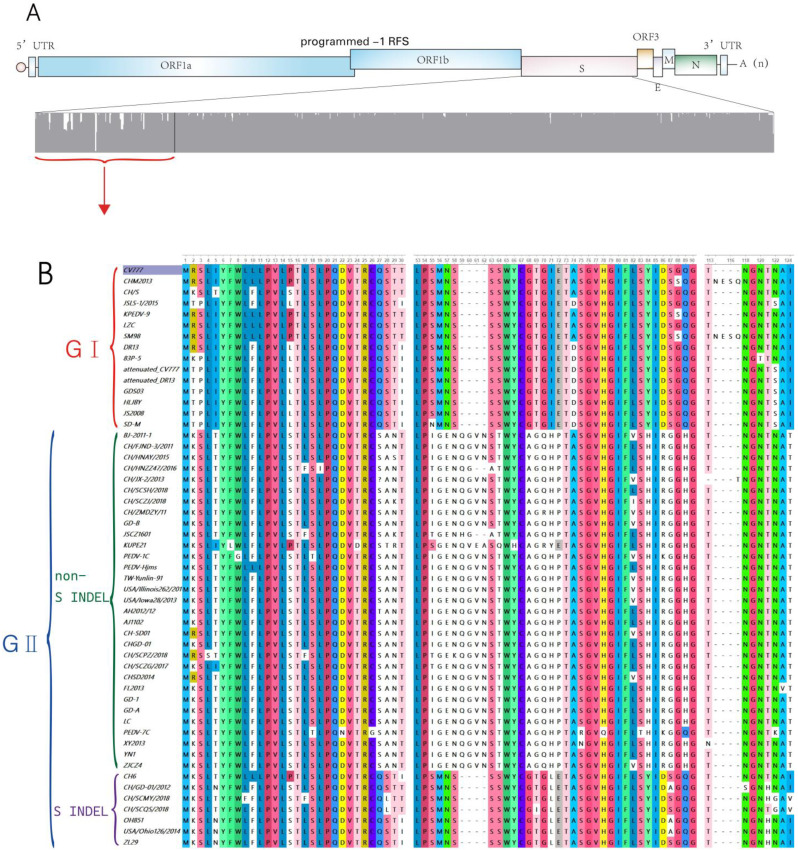
(**A**) The amino acid sequence differences of S proteins of different PEDV strains are shown, with the differences mainly concentrated in the N-terminal of S protein. (**B**,**C**) Amino acid sequences of the highly variable N-terminal region of the S proteins of various PEDV strains are shown in detail; the different genetic subgroups are indicated by brackets of different colors. The G2 non-S INDEL PEDV strain has a different genetic profile than CV777, while the G2 S INDEL PEDV strain has a higher similarity to CV777.

**Figure 3 viruses-14-01744-f003:**
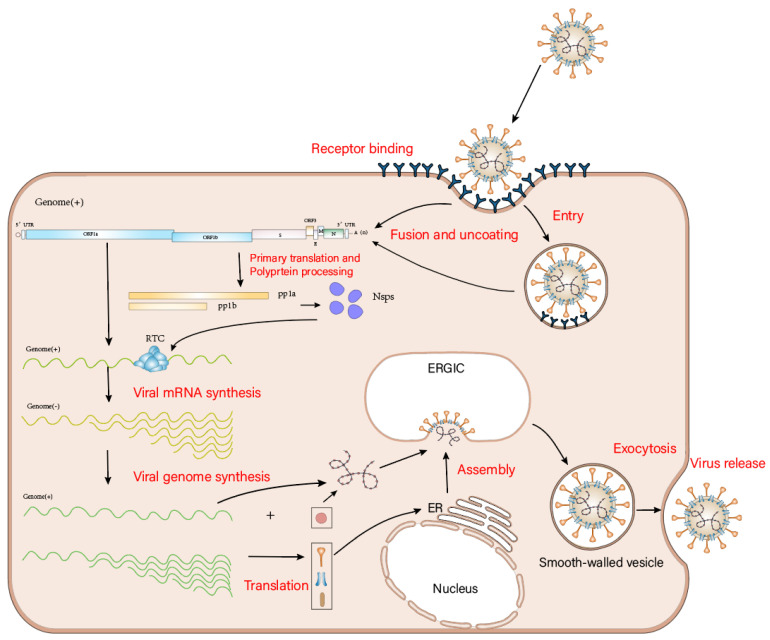
Overview of the PEDV life cycle.

**Figure 4 viruses-14-01744-f004:**
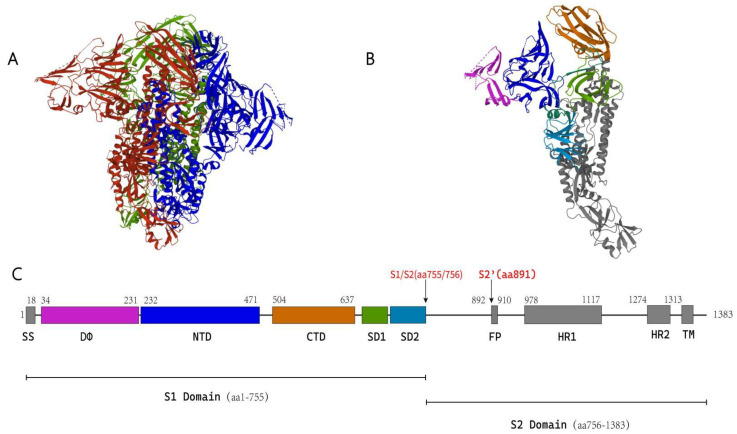
(**A**) The trimeric pre-fusion structure of the PEDV S protein is shown, with different colors rep-resenting different monomers. (**B**) Showing the structure of a monomer of the PEDV S protein. (**C**) Schematic diagram of the structure of a monomer of the PEDV S protein. Different colored boxes are used to represent the different structural domains of PEDV S proteins; the same color in di-agrams (**B**,**C**) indicates the same structural domain.

**Table 1 viruses-14-01744-t001:** Similarity in the amino acid residues of the S protein of different PEDV strains (CV777 was used as a reference).

Subgroup		Strain	Number of AA Dissimilarities	AA Similarity (%)	GeneBank
GⅠ		CV777	0	100	AF353511
		CHM2013	25	98	KM887144
		CH/S	53	96	JN547228
		JSLS-1/2015	59	96	KX534205
		KPEDV-9	42	97	KF898124
		LZC	13	99	EF185992
		SM98	24	98	GU937797
		DR13	28	98	JQ023161
		83P-5	46	97	AB548618
		attenuated CV777	57	96	JN599150
		attenuated DR13	54	96	JQ023162
		GDS03	59	96	AB857235
		HLJBY	58	96	KP403802
		JS2008	55	96	KC210146
		SD-M	59	96	JX560761
GⅡ	non-S INDEL	BJ-2011-1	92	93	JN825712
		CH/FJND-3/2011	95	93	JQ282909
		CH/HNAY/2015	96	93	KR809885
		CH/HNZZ47/2016	97	93	KX981440
		CH/JX-2/2013	104	93	KJ526096
		CH/SCSH/2018	94	93	MH053413
		CH/SCZJ/2018	101	93	MH061342
		CH/ZMDZY/11	98	93	KC196276
		GD-B	95	93	JX088695
		JSCZ1601	97	93	KY070587
		KUPE21	89	94	MF737355
		PEDV-1C	142	90	KM609203
		PEDV-Hjms	100	93	KY007139
		TW-Yunlin-91	96	93	KP276248
		USA/Illinois262/2014	94	93	KR265796
		USA/Iowa28/2013	95	93	KJ645636
		AH2012/12	100	93	KU646831
		AJ1102	95	93	JX188454
		CH-SD01	96	93	KU380331
		CHGD-01	96	93	JX261936
		CH/SCPZ/2018	98	93	MH593147
		CH/SCZG/2017	95	93	MH061337
		CHSD2014	96	93	KX791060
		FL2013	113	93	KP765609
		GD-1	99	93	JX647847
		GD-A	97	93	JX112709
		LC	95	93	JX489155
		PEDV-7C	135	90	KM609204
		XY2013	102	93	KR818832
		YN1	97	93	KT021227
		ZJCZ4	95	93	JX524137
	S INDEL	CH6	57	96	JQ239434
		CH/GD-01/2012	59	96	KP870113
		CH/SCMY/2018	71	95	MH061343
		CH/SCQS/2018	58	96	MH593141
		OH851	53	96	KJ399978
		USA/Ohio126/2014	51	96	KJ645702
		ZL29	55	96	KU847996

**Table 2 viruses-14-01744-t002:** Positions of the RBDs of some coronavirus binding receptors.

Virus	Receptor	Genus	RBD Position	Reference
PEDV	-	Alphacoronavirus	aa 477–629	[57]
TGEV	pAPN	Alphacoronavirus	aa 506–655	[53]
hCoV-NL63	ACE2	Alphacoronavirus	aa 476–616	[53]
hCoV-229E	hAPN	Alphacoronavirus	aa 293–435	[58]
MHV	mCEACAM1a	Betacoronavirus group A	aa 15–268	[59]
hCoV-HKU1	-	Betacoronavirus group A	aa 535–673	[60]
SARS-CoV	ACE2	Betacoronavirus group B	aa 318–510	[52]
SARS-CoV-2	ACE2	Betacoronavirus group B	aa 331–528	[38]
MERS-CoV	DPP4	Betacoronavirus group C	aa 318–588	[61]
PDCoV	pAPN	Deltacoronavirus	aa 302–425	[62]

## Data Availability

Not applicable.

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
