# Peer review of "PEDV: Insights and Advances into Types, Function, Structure, and Receptor Recognition"

_viruses, 2022, doi:10.3390/v14081744_

Round 1
Reviewer 1 Report
The overall logic and flow of the paper look fine. Authors did put a lot of effort working on it. This is a paper with tons of information, which is nice, and readers should learn a lot from reading it. I still believe the topic of the paper is significant, since PEDV is also a coronavirus and people are still being impacted by COVID, and pig industry is closely related to human world, including economics, society, public health and so on. Generally speaking, understanding PEDV is very necessary. Some points still need to be changed:
1. Line 28 subtitle: The structure and genome of PEDV. Based on the contents under this subtitle, I would not call it as “the structure” since you only described about the genome, the proteins and protein functions and you described structures in the later sections. I would just say something like “an overview of the genome and protein of PEDV”.
2. For the cryo-EM structure, if possible, I would suggest adapting the atomic level figures. So, when you describe your proteins, like the atomic structure of the S protein, it would be nice for readers to visualize the structure while reading your texts. A structure would be nicer adding to what you have now in fig.4, which currently is about the components of domains.
3. Below line 156, that’s the structure part. Slightly confusing since it’s kind of jumping back and forth from PEDV to general Coronavirus. You can simply describe PEDV protein’s structure and mention what else is known on this protein for other related viruses, or describe what is known for general Coronaviruses, then what is similar or special for PEDV. Just make the structure section more organized.
4. Please check the formatting of the paper. Super confusing while reading the texts. The subtitles and fonts need to be arranged.
Author Response
Response to Reviewer 1 Comments
Dear Editor and Reviewers:
Thank you very much for your positive and constructive comments on our manuscript entitled Advances in the study of porcine epidemic diarrhea virus (PEDV) entry into host cells (ID: viruses-1829229)
We have carefully studied the reviewers' comments and have done our best to revise our manuscript based on them. Below are our responses and revisions for each of the reviewers' questions and suggestions. Thanks again to the editors and reviewers for their hard work and valuable comments!
Response to the comments of Reviewer 1
Comment No. 1:Line 28 subtitle: The structure and genome of PEDV. Based on the contents under this subtitle, I would not call it as “the structure” since you only described about the genome, the proteins and protein functions and you described structures in the later sections. I would just say something like “an overview of the genome and protein of PEDV”.
Response: Thanks to the reviewer's reminder, We have replaced the subtitle “The structure and genome of PEDV” with“an overview of the genome and protein of PEDV”
Comment No. 2:For the cryo-EM structure, if possible, I would suggest adapting the atomic level figures. So, when you describe your proteins, like the atomic structure of the S protein, it would be nice for readers to visualize the structure while reading your texts. A structure would be nicer adding to what you have now in fig.4, which currently is about the components of domains.
Response: In response to the authors' comments,In fig.4, we added the Prefusion structure figures of PEDV spike cryo-electron microscopy, including trimer and monomer
Comment No. 3:Below line 156, that’s the structure part. Slightly confusing since it’s kind of jumping back and forth from PEDV to general Coronavirus. You can simply describe PEDV protein’s structure and mention what else is known on this protein for other related viruses, or describe what is known for general Coronaviruses, then what is similar or special for PEDV. Just make the structure section more organized.
Response: In response to the unreasonable narrative style of the structural part, we adjusted the introduction order and narrative style of general Coronavirus and PEDV, trying to make the structure more reasonable and easier to understand.
Comment No. 4:Please check the formatting of the paper. Super confusing while reading the texts. The subtitles and fonts need to be arranged.
Response: We checked the subtitles and fonts of the article, and revised the language of the whole article, hoping to improve the overall quality of the article
Special thanks to you for your good comments
We tried our best to improve improve the manuscript,and made some changes in the manuscript. And here we marked in red in revised paper.
We appreciate for Editors/Reviewers’ warmwork earnestly, and hope that the correction will meet with approval. Once again, thank you very much for your comments and suggestions
Yours sincerely
Feng Lin
Correspondence to: Xiaochun Tang, xiaochuntang@jlu.edu.cn
Reviewer 2 Report
This review summarized the recent study of PEDV entry, including viral structure, types of PEDV strains, life cycle of PEDV, S protein and advances in PEDV receptor research. The review contains a lot of information. While the main topic as shown in the title is PEDV entry into host cells, the first three parts (viral structure, types of PEDV strains, life cycle of PEDV) make up almost half of the main context, these parts should be briefly introduced and highlight the points which are related to PEDV cell entry. And English polishing service is suggested to improve the overall quality of the manuscript. Please see following some comments.
Some of the titles of the paragraphs are in bold and some are not, it causes a little bit confusion. It’s better to use subheadings if necessary.
Line35 “Nonstructural proteins (nsps), ORF1a, ORF1b, and ORF3” should be changed to “pp1a, pp1ab and ORF3 protein” (ORF1a and ORF1b are open reading frames not proteins. NSPs are not directly encoded by the ORFs, pp1a, pp1ab are post-processed to generate the NSPs.)
Line 163-165 Please double check if FP and HR region are located at the N-terminus of the viral fusion protein.
Author Response
Response to Reviewer 2 Comments
Dear Editor and Reviewers:
Thank you very much for your positive and constructive comments on our manuscript entitled Advances in the study of porcine epidemic diarrhea virus (PEDV) entry into host cells (ID: viruses-1829229)
We have carefully studied the reviewers' comments and have done our best to revise our manuscript based on them. Below are our responses and revisions for each of the reviewers' questions and suggestions. Thanks again to the editors and reviewers for their hard work and valuable comments!
Response to the comments of Reviewer 2
Comment No. 1:Some of the titles of the paragraphs are in bold and some are not, it causes a little bit confusion. It’s better to use subheadings if necessary.
Response: Thanks to the reviewer's reminder, we adjusted the title of the whole paper and added some subheadings.
Comment No. 2:Line35 “Nonstructural proteins (nsps), ORF1a, ORF1b, and ORF3” should be changed to “pp1a, pp1ab and ORF3 protein” (ORF1a and ORF1b are open reading frames not proteins. NSPs are not directly encoded by the ORFs, pp1a, pp1ab are post-processed to generate the NSPs.)
Response: In response to the reviewer's comments Line35 "Nonstructural proteins (nsps), ORF1a, ORF1b, and ORF3" has been changed to "pp1a, pp1ab and ORF3 protein"
Comment No. 3:Line 163-165 Please double check if FP and HR region are located at the N-terminus of the viral fusion protein.
Response: We apologize for the trouble our misrepresentation caused to the reviewers and have changed the original text to "type I viral fusion proteins contain a heptad repeat (HR) region and an S2 subunit N-terminal or N-proximal fusion peptide(FP)".
Special thanks to you for your good comments
We tried our best to improve improve the manuscript,and made some changes in the manuscript. And here we marked in red in revised paper.
We appreciate for Editors/Reviewers’ warmwork earnestly, and hope that the correction will meet with approval. Once again, thank you very much for your comments and suggestions
Yours sincerely
Feng Lin
Correspondence to: Xiaochun Tang, xiaochuntang@jlu.edu.cn
Round 2
Reviewer 2 Report
The overall quality of the manuscript has been improved. One thing is that the sequence alignment figure is not clear and hard to read.
Author Response
Response to Reviewer 2 Comments
Dear Editor and Reviewers:
Thank you very much for your positive and constructive comments(Round 2) on our manuscript entitled PEDV:Insights and advances into types,function,structure, and receptor recognition (ID: viruses-1829229)
We have carefully studied the reviewers' comments and have done our best to revise our manuscript based on them. Thanks again to the editors and reviewers for their hard work and valuable comments!
Response to the comments of Reviewer 2
Comment :The overall quality of the manuscript has been improved. One thing is that the sequence alignment figure is not clear and hard to read.
Response: Thanks to the reviewers' comment, we have revised the sequence alignment figure to ensure that it is clear and readable.
In addition, we embellished and revised the language of the manuscript to improve the overall quality of the manuscript and make it more readable
We appreciate for Editors/Reviewers’ warmwork earnestly, and hope that the correction will meet with approval. Once again, thank you very much for your comments and suggestions
Yours sincerely
Feng Lin
Correspondence to: Xiaochun Tang, xiaochuntang@jlu.edu.cn

This manuscript is a resubmission of an earlier submission. The following is a list of the peer review reports and author responses from that submission.
Round 1
Reviewer 1 Report
The authors talked about the entry of PEDV from the aspects of the genome the viral proteins encoded, different type of strains, the mutations and where were the strains found, the replication cycle, the structural and functional details of the spike protein, and different PEDV receptor studies. This study has a big significance since PEDV is related to SARS-CoV and the outbreaks of PEDV will bring a serious impact in human economy. Overall, the article is rich in content and organized well. Please check the following comments:
1. References. The citations are good, however, usually we would like to see citations after each solid statement you made in a sentence. So, a lot of places lack of citations. Please check your entire paper and add citations accordingly.
2. Is there an atomic level structure of this entire virus been solved? If not, why there’s not yet a cryo-EM structure? Is there a bottleneck? What are the closest viruses with a whole atomic structure? This could be mentioned in the beginning when you show the cartoon of the virus.
Reviewer 2 Report
In this review, Feng Lin et al. went through the previous and recent study of PEDV cell entry, focusing on the spike protein and its receptor. While the paper contains a lot of information, it was not written very well and a little tedious. Some of the expressions and statements were improper and a little bit hard to follow. Extensive English language and style editing are suggested. And it’s better to have the number of each line labeled in the manuscript for review. Some other suggestions are as follows.
Figure 1. Coronavirus ORF1a and ORF1b usually has an overlap region where the programmed –1 ribosomal frameshift happens. So, when drawing the ORF1a and ORF1b, one needs to pay attention to the overlap at the adjacent area within the two ORFs.
Figure 2. The replication machinery of coronavirus contains a lot of proteins except for the RdRp, like nsp7, nsp8, nsp9, nsp10, nsp13 and nsp14… And it is usually named as RTC, using only RdRp is not accurate.
Coronavirus is proved to have their viral genomic RNA replication and subgenomic mRNAs transcription happened in the double-membrane vesicles (DMVs), not in the cytoplasm.
While the authors were talking about the residue differences of different PEDV strains or at other part, it will be good to have a sequence alignment.